# Application of Biomass Materials in Zinc-Ion Batteries

**DOI:** 10.3390/molecules28062436

**Published:** 2023-03-07

**Authors:** Yu Zhang, Mengdie Xu, Xin Jia, Fangjun Liu, Junlong Yao, Ruofei Hu, Xueliang Jiang, Peng Yu, Huan Yang

**Affiliations:** 1Hubei Key Laboratory of Plasma Chemistry and Advanced Materials, School of Materials Science and Engineering, Wuhan Institute of Technology, Wuhan 430205, China; 2Department of Food Science and Chemical Engineering, Hubei University of Arts and Science, Xianyang 441053, China; 3Hubei Key Laboratory of Polymer Materials, Hubei University, Wuhan 430205, China

**Keywords:** biomass, zinc-ion battery, electrode, electrolyte, separator, binder

## Abstract

Currently, aqueous zinc-ion batteries, with large reserves of zinc metal and maturity of production, are a promising alternative to sustainable energy storage. Nevertheless, aqueous solution has poor frost resistance and is prone to side reactions. In addition, zinc dendrites also limit the performance of zinc-ion batteries. Biomass, with complex molecular structure and abundant functional groups, makes it have great application prospects. In this review, the research progress of biomass and its derived materials used in zinc-ion batteries are reviewed. The different regulation strategies and characteristics of biomass used in zinc-ion battery electrodes, electrolyte separators and binders are demonstrated. The regulation mechanism is analyzed. At the end, the development prospect and challenges of biomass in energy materials application are proposed.

## 1. Introduction

In order to realize the combination of world development and environmental protection, renewable energy is widely popularized [1]. Aqueous zinc-ion batteries, with high safety, non-toxicity and ease of manufacture, have attracted great research interest. Zinc metal is stable in the natural environment and can be directly used to assemble batteries without being assembled in a harsh glove box, which greatly reduces the production cost and production conditions of zinc-ion batteries. Moreover, it uses water as an electrolyte, which is environmentally friendly and safe compared with organic electrolytes [2,3]. However, with the study of zinc-ion batteries, the generation of zinc dendrites, the dissolution of the electrode, the occurrence of side reactions at the electrolyte and electrode-electrolyte interface, the ion transport properties of the electrolyte and the separator, and the viscosity and flexibility of the binder are all problems that limit the development of zinc-ion batteries.

Biomass, with unique structural and chemical properties, is a kind of renewable organic material [4]. Biomass feedstocks, such as wood, crab shell, and waste, etc., can be used to produce porous carbon through physical/chemical activation, these as-prepared carbon can be used as electrode material for ZIBs [2]. Furthermore, many pure biomass components, such as cellulose, lignin, chitin/chitosan, etc., can be exacted from biomass feedstocks. For example, cellulose can be extracted from terrestrial and aquatic plant species by using a top-down mechanically/chemically induced extraction strategy. These pure biomass components, with well-defined chemical structure, have a broad and promising application in ZIBs. Biomass materials can be assembled into various nano-scale devices to meet the design requirements of zinc ion energy storage equipment [5]. For example, activated carbon with porous structure can be obtained by carbonization of straw biomass waste [6]. As an electrode or electrode coating, it can improve the active sites on the electrode surface, and effectively inhibit the dissolution of the electrode and the generation of zinc dendrites. Hydrophilic biomass polymers, such as gelatin, agar and cellulose [7] can be used to prepare gel or solid electrolyte, which effectively prevents the occurrence of side reactions in aqueous electrolyte [8]. However, the frost resistance of the gel itself and the poor conductivity of the biomass need to be improved by introducing other conductive antifreeze compounds. Biomass, such as lignin and gelatin, can be introduced by solution casting [9]. The large number of micropores and tough network structures on biomass can increase the mechanical properties of the diaphragm and the ion exchange rate. Biomass with excellent adhesion and flexibility can be directly used as binder. Although it provides better adhesion, the uneven distribution of biomass macromolecules will increase the resistance of the battery, which requires more detailed regulation of the distribution of biomass macromolecules [10,11].

In this review, we start with different components of zinc-ion batteries and comprehensively summarize some problems in batteries and propose strategies to solve these problems by using biomass with different components. The application of biomass in zinc-ion battery devices, electrodes, electrolytes, separators and binders are reviewed. However, there are still many difficulties to be solved to realize the industrial application of biomass in zinc-ion batteries. It is necessary to realize the perfect combination of biomass and existing commercial materials or to develop innovative applications of biomass. In addition, this review also expounds the existing limitations and future development of zinc-ion batteries, which provides a reference for the rational design and utilization of biomass and its derived materials to prepare zinc-ion batteries with long service life and high energy density and provides a development direction for future zinc-ion batteries and new energy storage.

## 2. Zinc-Ion Battery

Zinc-ion battery is mainly composed of positive and negative electrode materials, electrolyte, separator and binder. The reversible zinc stripping/electroplating of the negative electrode and the reversible Zn^2+^ insertion/extraction of the positive electrode realize the energy storage and release of the zinc-ion battery [12]. The electrolyte transmits the transport current to the ion medium, and its ionic conductivity and operating temperature range affect the performance and operating temperature of the entire battery [13,14]. The separator acts as a physical barrier between the electrodes to prevent internal short circuits, which is a key factor in determining the safety of zinc ion energy storage devices [15,16]. The binder is responsible for connecting the electrode materials together and then stably adhering them to the collector to provide good contact for the electron flow [17,18]. The charge storage process of a zinc-ion battery depends on the migration of Zn^2+^ between the anode and cathode. The charge storage mechanisms of Zn anode and cathode of a zinc-ion battery are briefly discussed as follows:

### 2.1. Charge Storage Mechanism of Zn Anode

The charge storage of zinc anode relies on the reversible zinc stripping/electroplating. As shown in Equation (1).
(1)$${\mathrm{Zn}}^{2+}+2e^{-}⇌\mathrm{Zn}$$

During the charging process of ZIB, Zn^2+^ in the electrolyte is reduced and deposited on the Zn anode. During the discharge process, the anode zinc is stripped and oxidized to soluble Zn^2+^, which migrates to the cathode side [19,20]. However, Zn^2+^ is more inclined to deposit on the thread dislocation on the surface of Zn, and uneven stripping/plating leads to a series of side reactions such as zinc dendrites [19,21,22].

### 2.2. Charge Storage Mechanism of Cathode

At present, the energy storage mechanism of different cathode materials can be divided into four categories: traditional Zn^2+^ insertion/extraction, dual ions co-insertion, Zn^2+^ coordination/uncoordination and conversion reaction [12,19].

(1) Traditional Zn^2+^ insertion/extraction

Reversible Zn^2+^ insertion/extraction has been widely accepted as the main mechanism of most ZIB cathodes, which is suitable for cathodes with open structures, such as special channels and layered structures [19,23]. Taking the classical α-MnO_2_ tunnel structure as an example (Equation (2)), Zn^2+^ ions are extracted from the tunnel of α-MnO_2_ during charging, and Zn^2+^ ions dissolved from the Zn anode move rapidly and insert into the tunnel during discharging [20].
(2)$${\mathrm{Zn}}^{2+}+2e^{-}+2{\mathrm{MnO}}_{2}⇌{\mathrm{ZnMn}}_{2}O_{4}$$

(2) Dual ions co-insertion

The large size of water and Zn^2+^ will produce strong electrostatic repulsion with other divalent ions, resulting in slow Zn^2+^ intercalation. Different from traditional Zn^2+^ insertion/extraction, the dual ions co-insertion mechanism is also demonstrated. In theory, the insertion of intercalation ions such as H^+^, Li^+^, and Na^+^ with faster diffusion is allowed at the same time as the insertion of the host material, which effectively promotes the use of active sites in the host structure and improves the performance of zinc-ion batteries [12,24,25].

(3) Coordination/uncoordination reaction of Zn^2+^ with organic cathode

This principle is suitable for organic compound cathode materials with abundant electroactive groups, which act as excellent active sites for Zn^2+^ storage by rapid coordination with electropositive Zn^2+^ [19,20]. These organic cathodes include quinone compound [26], 2D covalent organic frameworks [27], metal-organic framework [28] and pyrene-4,5,9,10-tetraone [29], etc.

(4) Conversion reaction

The conversion reaction in the battery can directly carry out charge transfer. A reasonable conversion reaction can improve the performance of zinc-ion battery. There is a reversible conversion between α-MnO_2_ and MnOOH in a mild ZnSO_4_ aqueous electrolyte during the charging/discharging process of Zn/α-MnO_2_ batteries [30].

The energy density of zinc-ion batteries varies from tens to few hundreds Wh kg^−1^. For instance, the zinc-ion battery with the modulated NiCO_2_O_4_ nanosheets as the cathode can reach the energy density of 578.1 Wh kg^−1^ [31]. The development of zinc-ion batteries with high energy density is also one of the reasons for applying biomass to batteries.

## 3. Biomass for Electrodes

In general, the cathode materials of zinc-ion batteries include manganese-based oxides [30,32,33] as well as the prussian blue analogues [34,35], inorganic molybdenum sulfate [36], molybdenum oxide [8,37] and organic quinone compounds, which exhibit tunnel structure or layered structure. Zinc ions can be reversibly embedded and extracted in the cathode material. However, due to the tip effect, the microscopic protrusions on the surface of the metal zinc anode produce a strong local electric field during charging, attracting Zn^2+^ to deposit and grow into large-size zinc dendrites [38]. Biomass can enrich the active sites of electrode materials, expand the space for storing ions and electrons on the surface of electrode materials, and effectively solve the problem of zinc dendrites. Currently, a variety of biomass materials, such as yeast [39], collagen [40], gelatin and agar [41], and even biomass waste, can be used as electrode materials for zinc-ion batteries. In addition, organic polymer biomass, the coating of hydrophilic biomass, and the 3D modification of carbonized biomass are typical advanced biomass for electrode materials [42,43].

### 3.1. Pure Biomass for Organic Electrode Materials

Compared with conventional inorganic cathode materials, the organic cathode materials have the following advantages, such as environmentally friendly and renewable [44,45,46], adjustable structure [47], high charge and discharge rate. Generally, organic cathode materials only involve chemical absorption and ion doping during charging/discharging process, which cannot change their structure and valence bond, resulting in high charge and discharge rate [23], Quinones present completely insoluble in water, which have a variety of properties when applied to zinc-ion batteries [48]. Quinone organic materials have weak intermolecular van der Waals forces, which can produce moderate Coulomb repulsion to diffuse cations [49,50,51]. Moreover, their plasticity and soft lattice may allow molecules to reorient, which are conducive to form the reversible intercalation of Zn^2+^. Therefore, the carbonyl reduction potential can be adjusted by modifying the function of quinone derivatives (The electrode reaction of the quinone electrode is shown in Figure 1a) [52,53,54]. For example, the p-aminobenzoic acid is synthesized by using yeast and wheat bran as precursors (Figure 1b), which presents a cavity structure. The carbonyl groups on and under the calix[4] quinone molecule with an open bowl structure and eight carbonyl groups can react with Zn^2+^, which is more conducive to the absorption of Zn^2+^ (Figure 1c,d), increasing the capacity and energy density of the battery [39]. At present, more than 2400 quinone compounds have been extracted from plants, animals, fungi and other biomass [55]. However, Zn anode is inevitably poisoned by adopting quinone compounds in aqueous solution. In this regard, ion exchange membrane is used to effectively inhibit the dissolution of discharge products [49,56].

### 3.2. Biomass with Surface Modification for Electrode Materials

Compared with common carbon-based coatings, biomass coatings have certain water absorption. For example, the coverage of cathode coatings, such as gelatin and agar, can reduce the interface impedance and increase the diffusion rate of Zn^2+^ on the electrode, so as to improve the capacity and cycle life of zinc-ion battery [57,58]. Furthermore, the coating can also act as a barrier on the electrode surface, physically inhibit the dissolution of the electrode and maintain the stability of the electrode structure [57]. In addition, the functional groups, such as carboxyl groups contained in the biomass coating, can also effectively adsorb and limit the dissolved Mn^2+^ and Zn^2+^ on the electrode surface, achieving a high capacity beyond the intercalation capacity [40,59]. For instance, collagen can also form micro-skin on the electrode surface after hydrolysis, and adsorb dissolved Mn^2+^, thus improving the reversibility of the total redox reaction of MnO_2_/Mn^2+^ during charge and discharge, improving the cycle life of zinc-ion batteries [40].

The surface modification of the cathode material can slow down the occurrence of side reactions on the EEI effectively; thus, the introduced coating on the surface of the zinc anode can achieve a dendrite-free zinc anode, which is an effective strategy to induce uniform deposition of zinc [60]. MXene films prepared with chitosan can be used as coatings for zinc anodes [38]. This amine-rich ultra-thin protective film not only enhances the hydrophilicity of the anode, but also effectively prevents the corrosion of water molecules on the zinc anode on the basis of ensuring the transport of zinc ions and promotes the uniform nucleation of zinc (Figure 1e) [38]. Moreover, the volume change of zinc metal anode during cycling is buffered, which makes the MXene/chitosan @Zn anode exhibit a longer cycle life and high reversibility [38]. The coating thickness also has an effect on the performance of zinc-ion battery. The cycling stability of the electrode coated with 1% agar solution is higher than that of MnO_2_ electrode without agar solution and the MnO_2_ electrode coated with 2.5% agar solution (Figure 1f) [57]. The thin coating can decrease the charge transfer resistance and improve the reversibility of the MnO_2_/Mn^2+^ reaction. Nevertheless, the thick micro-skin coating increases the interface resistance and hinders the transmission speed of Zn^2+^, reducing the specific capacity of battery. Therefore, coating can avoid direct contact with the electrolyte and reduce the occurrence of various corrosion and side reactions, regulating the coating thickness can effectively improve the performance of the battery [61].

### 3.3. Biomass Derived Carbon for Electrode Materials

Carbon materials usually have high conductivity, good chemical stability and rich pore structure, which can be combined with cathode-active materials to effectively improve their performance [62]. In general, activated carbon materials with rich porous structure can be obtained by biomass carbonization, the induced high porosity can promote the rapid transfer of electrons on composite materials (Figure 2a) [62]. For instance, carbon materials are prepared by chemical activation, and Mn_3_O_4_ particles are coated by deposition (Figure 2b) [62]. This obtained with an additional mesoporous structure, act as an environmentally friendly and efficient manganese oxide support for zinc-ion batteries [62]. A three-dimensional (3D) carbon material (COG), obtained by high-temperature carbonization of marine waste-reed straw, exhibits a honeycomb cell structure with multi-level open channels [63]. Afterwards, nano sheets of Zn and MnO_2_ are deposited on the surface of multi-COG to obtain COG@Zn and COG@MnO_2_, which protect the zinc anode and achieves high-quality load of the electrode (Figure 2c) [63]. The conductive porous carbon host material ensures rapid electron transfer, reduces local current density, alleviates tip effect, reduces zinc nucleation overpotential, and achieves highly reversible and uniform zinc deposition [62]. In addition, the porous carbon layer provides abundant ion channels to guide the migration path of Zn^2+^ and adjust the Zn^2+^ nucleation sites, achieving high-quality loading and providing a low-resistance path [64].

The doping of heteroatoms such as N can effectively increase the surface roughness of biomass materials, increase their surface area and adjust the hydrophilicity of materials. In general, N atoms can exist in biomass-derived carbon skeletons in various forms, such as C-N, C=N, N-O, N-H, graphite N, pyridine- (N^+^-O^-^), pyrrole N, and pyridine N [65,66]. For instance, porous carbon is constructed by activation of N-doped corn silk, which is introduced into the zinc-based metal organic framework (Figure 3a) [67]. The N doping endows ZnO/N/C with a hierarchical porous structure (Figure 3b) [67]. The interconnection of macropores and mesopores can achieve rapid transport of zinc ions, which is of great significance for obtaining a high-rate capability [68]. The zinc-ion battery with ZnO/N/C as the cathode shows a capacity retention of 97.0% after 8000 charge–discharge cycles at 1.0 A g^−1^ (Figure 3c) [67].

## 4. Biomass Used in Electrolytes

Biomass with hydrophilic functional groups, such as NH_2_, -OH, -CONH-, -CONH_2_ and -SO_3_H, have high adsorption affinity for polar solvent molecules. Its strong interaction with salt anions can enhance the salt solubility and cation transport properties, promote the distribution of Zn^2+^ on the electrode surface more uniformly, thereby inhibiting the growth of Zn^2+^ crystals [69]. Biomass polymers, such as cellulose [70], guar gum [8], xanthan gum [71], carrageenan [56], sodium alginate [72], silk fibroin [73], can be used as electrolytes. Their mechanical stiffness and flexible combination can adapt to the changed electrode volume, resulting in a stable interface and the extended service life [74,75,76]. They can also be processed into customized shapes with uniform nanopore distribution to form a uniform metal ion flux at the electrode–electrolyte interface, which is conducive to providing stable zinc deposition/stripping, thereby effectively avoiding the generation of zinc dendrites [77].

### 4.1. Single Biomass Electrolyte

Single macromolecular biomass can coat on the target substrate to construct a solid-state zinc-ion battery, which can be beneficial to achieve a good bonding interface for charge and mass transfer, slowing down self-corrosion and eliminating a series of side reactions in aqueous solution (Figure 4a) [8,78]. Biomass, such as guar gum, gelatin and xanthan gum, can be used as electrolytes for zinc-ion batteries (Figure 4b,c). For instance, the ionic conductivity of the salt-tolerant gel electrolyte (20 wt% xanthan gum/3 M ZnSO_4_/0.1 M MnSO_4_) is 1.65 × 10^−2^ S cm^−1^, which is much higher than other polymer electrolytes (Figure 4c) [71]. In addition, the conductivity of the electrolyte shows a certain frost resistance and long-term stability [71]. Compared with xanthan gum electrolyte, guar gum electrolyte has a wide temperature window (5~45 °C) and higher conductivity, which can be up to 4.8 × 10^−2^ S cm^−1^ [8]. Moreover, the battery with guar gum as an electrolyte has a more excellent power density and rate performance (181.6 mAh g^−1^, 3 A g^−1^) (Figure 4d) [8].

Cellulose is an excellent biopolymer electrolyte for zinc-ion batteries, which can effectively alleviate the problem that gel electrolyte is difficult to adapt to rough zinc anode [77]. A viscoplastic gel electrolyte is prepared with an interfacial adaptability by using cellulose as a precursor, which can optimize the contact interface between the electrode and the electrolyte, guiding the homogeneous epitaxial uniform deposition of zinc by adjusting the solvation structure of Zn^2+^ (Figure 5a) [79]. Compared with the conventional single strategy for regulating the nucleation of zinc crystals in the electrochemical process, the functionalized flexible gel electrolyte can achieve the stability of the zinc anode. The charged groups in the gel electrolyte construct a channel for the efficient movement of Zn^2+^ and homogenize the interfacial electric field of the zinc anode, which can realize the preferential growth of the zinc crystal plane (002) and optimize the deposition kinetics of Zn^2+^ (Figure 5b) [7,80,81,82]. Single biomass as gel electrolyte can effectively solve the side reactions in aqueous solution and insufficient interface adaptability of aqueous zinc-ion batteries. However, their poor frost resistance still exists and needs to be solved.

### 4.2. Biomass Mixed Electrolyte

Biomass has upgraded the electrolyte of the zinc-ion battery from an aqueous solution system to a hydrogel or solid film package. However, the system with a large amount of water makes the battery perform poorly at low temperatures. Introducing ethylene glycol, Zn(ClO_4_)_2_ and other substances into the biomass electrolyte can develop a zinc-ion battery with antifreeze performance [17,83,84]. For instance, the hydrogel electrolyte is obtained by introducing ethylene glycol into the guar gum/sodium alginate electrolyte, which presents excellent antifreeze properties (Figure 6a) [17,85]. Guar gum can enhance the mechanical strength and toughness of the hydrogel electrolyte. In addition, due to the abundant hydrophilic functional groups and carboxyl groups, sodium alginate improves the ionic conductivity of the gel electrolyte at room temperature (Figure 6b) [85]. Furthermore, ethylene glycol and Zn(ClO_4_)_2_ can be introduced to improve the antifreeze performance of biomass electrolytes, which can be ascribed to prevent the formation of ordered hydrogen bond networks between water molecules, reducing the freezing point of water electrolytes and improving the antifreeze performance of batteries (Figure 6b–d) [85,86]. Compared with the zero conductivity at −20 °C of guar gum, guar gum/sodium alginate electrolyte, guar gum/sodium alginate/ethylene glycol hydrogel electrolyte still maintains the conductivity of 6~7 Ms cm^−1^ (Figure 6b). More importantly, the discharge capacity can still provide more than 100 mAh g^−1^ at −20 °C (Figure 6c) [85].

Combining biomass gel electrolyte with antifreeze is an effective strategy to improve the antifreeze performance of the battery. For example, adding calcium and gelatin to silk fibroin is applied to prepare a humidity-sensitive plasticized gelatin-silk fibroin electrolyte (Figure 7a), a fiber-type zinc-ion battery (TZIB) is construct by controlling the plasticization of it [73]. The gelatin-based three-dimensional network could effectively facilitate the Zn^2+^ transfer, reducing the energy loss during the ion migration process (Figure 7b) [73,87,88]. Moreover, gelatin has excellent temperature sensitivity and hydrophilicity. The absorption of water in the surrounding environment and the increase in temperature can make the gelatin liquefy, improve the corresponding efficiency of silk fibroin to water vapor, and realize rapid intelligent humidity plasticization (Figure 7c) [73]. Otherwise, the dried gelatin has excellent mechanical properties, which can effectively inhibit the growth rate of zinc dendrites, and has excellent waterproof and degradation properties [69,89]. Currently, the cross-used different biomass endows a zinc-ion battery electrolyte with many functions; nevertheless, most of the biomass has limited electrical conductivity and mechanical properties. Therefore, it is also necessary to effectively combine biomass with widely used polymers and other electrolytes to further improve its electrical conductivity and mechanical properties, realizing the real application of the biomass electrolyte.

### 4.3. Biomass-PAM Copolymer Electrolyte

Polyacrylamide (PAM) with certain electrical conductivity is widely used in water-soluble polymers. The active amide groups, anion and cation groups on the polymer chain can physically and chemically react with a variety of substances, enhancing the polymer framework and providing additional pore structure for the polymer [74]. Biomass with high electronegative groups can strongly form intermolecular hydrogen bonds with polymers [90,91]. Therefore, biomass and polyacrylamide can be used to prepare electrolytes, which present a certain degree of hierarchical porous network structure, good hydrophilicity, adhesion and flexibility [16,92].

The hierarchical porous network structure facilitates the capture of water in the network and the absorption of aqueous solutions, increasing the ionic conductivity by allowing free movement of Zn^2+^ ions. Based on the sodium alginate-polyacrylamide hydrogel electrolyte with a highly three-dimensional porous structure, zinc-ion batteries can be made more stable by combining a layered δ-Na_0.65_Mn_2_O_4_ 1.31H_2_O cathode material (Figure 8a) [93]. The PAM provides a kind of 3D porous structure for the layered hydrogel electrolyte SA-PAM, providing a large number of ion channels. Under the synergistic effect of various aspects, the capacity of δ-MnO_2_ almost reaches the theoretical value, which provides an excellent specific capacity for zinc-ion batteries. The strong contact at the interface greatly increases the ion migration rate and achieves rapid charge transfer [94]. Carboxymethyl cellulose [95], sodium lignosulfonate [5,96] can be copolymerized with acrylamide to obtain a biomass-PAM copolymer electrolyte (Figure 8b), which has excellent adhesion properties (Figure 8c) and can achieve a low electrode–electrolyte interface resistance and rapid ion conduction, acting as a separator to suppress the generation of zinc dendrites and maintain excellent capacity during severe bending deformation (Figure 8d) [93,96].

In addition, some biomass and PAM have hydrophobic and hydrophilic chain structures, their amphiphilic characteristics provide a high opportunity to form hydrogen bonds [97]. The increased hydrogen bonds of the polymer blends show miscibility, which increases the conductivity. Similar to the transport of Zn^2+^ in polymer electrolytes, the zinc ion–polymer complex in the salt jumps from one coordination site of the functional group to another coordination site [98,99]. The introduced biomass enriches the active sites of the electrolyte. Copolymerization of carrageenan with PAM yields a double-network hydrogel [56]. The main chain of the carrageenan polymer immobilizes anions, and the covalently cross-linked PAM network and the physically cross-linked carrageenan produce synergistic effects, resulting in a more flexible hydrogel system and achieving single zinc ion conduction (Figure 8e) [56,100,101]. This makes the electrolyte lack free anions, ensuring a uniform Zn^2+^ flux, greatly inhibiting the growth of zinc dendrites, and effectively preventing the passivation of zinc anodes [102,103,104].

## 5. Biomass for Separator

The diaphragm acts as a physical barrier between the electrodes, it can allow the electrolyte ions to move freely. The regulation of ion deposition can be achieved to avoid zinc dendrites [105]. Excellent performance of the separator can help to achieve high ion exchange rate, good mechanical properties and flexibility. Biopolymers contain a large number of complex functional groups, which can enhance the toughness of the separator, enrich the channels of water and ions, and increase electrical conductivity [9,17]. It can also cross link with the electrolyte and form hydrogen bonds with H_2_O, regulating the deposition of zinc anodes and inhibiting the dendritic failure in zinc-ion batteries [106]. At present, biomass as separators is mainly carried out by modifying separators and developing new biomass separators.

### 5.1. Biomass Modified Commercial Separator

As a widely used commercial separator, glass fiber has stable chemical properties, suitable ionic conductivity and good wettability. Nevertheless, it is easy for zinc dendrites with ultra-high Young’s modulus (108 GPa) to penetrate the separator [81,107]. Currently, biomass with functional groups can effectively modify the glass fiber separator. Gelatin is introduced into bare glass fiber to obtain a separator, which can inhibit the occurrence of a violent disproportionation reaction on the cathode material, thereby protecting its spinel structure [63]. Gelatin becomes a film connected to the glass fiber, increasing the toughness, flexibility and mechanical strength of the separator. In addition, gelatin can chelate Zn^2+^ into ZnGl^+^ during electroplating, which increases the charge transfer resistance and improves the capacity retention rate [108]. Furthermore, during the electroplating process, gelatin will adsorb on the surface of zinc and form a film, which can cover the active center on the battery electrode, reduce the nucleation rate and obtain a uniform dendrite-free coating and obtain a uniform dendritic-free coating [109]. Therefore, biomass can be modified to improve the mechanical properties of commercial separators and enhance the capacity retention.

### 5.2. Biomass Modified Nafion Membrane

The Nafion membrane is an excellent proton exchange membrane with stable performance and has an amazing long stripping/plating cycle life. Compared with commercial separators, the Nafion membrane has a better ion exchange performance. As shown in Figure 9a, it can provide binding sites for Zn^2+^ ions, effectively inhibiting the dissolution of organic cathode discharge products, and solve the exposition of zinc (100) [9]. Biomass modification can construct more conductive channels in the Nafion membrane, which can further improve the ion exchange performance. The lignin is introduced into the Nafion membrane by simple scaling after solution casting, the casting membrane immersed in ZnSO_4_ can convert the casting membrane into the Zn^2+^ type [94]. The obtained lignin@Nafion membrane not only has high conductivity, but also has a higher service life than that of commercial physical separators [9]. Nevertheless, the high price of a biomass-modified Nafion membrane makes zinc-ion batteries less attractive.

### 5.3. Biomass Directly Used as Separators

Metal-organic-framework-coated polyolefin separators, polyacrylonitrile and other new separators can also be used as ZIBs separators to inhibit the generation of zinc dendrites, but they are limited by the cumbersome synthesis process and environmental problems. Therefore, the commercial application of the pure biomass separator is more promising. Taking a cotton towel separator as an example, by comparing the pure biomass separator with a commercial glass fiber separator, the cotton towel separator has the advantages of rich porosity and low surface energy, which can effectively buffer hydrogen evolution and side reactions to improve the reversibility of the zinc anode [110,111]. In addition, the cellulose separator has a larger strength modulus, and its rich hydroxyl functional groups and uniform and dense nanopores make it have higher ionic conductivity. These can promote the migration of ions and charges, reduce the desolvation energy barrier of hydrated zinc ions, and accelerate the zinc deposition kinetics at the zinc electrode/electrolyte interface. Therefore, the cellulose membrane can effectively inhibit the occurrence of zinc dendrites and other harmful side reactions [112]. Nevertheless, the conductivity and porosity of the biomass itself limit its ion exchange rate, and biomass separators can be modified using ionic liquids to enhance their conductivity.

For batteries with solid electrolytes, the electrolyte also acts as a separator. For example, with carboxymethyl chitosan as a plasticizer and ionic liquid as a charge carrier, the obtained solid electrolyte can be used as separator in a zinc-ion battery. In this separator, the protonation of carboxymethyl chitosan treated by an acetic acid solution produces a positive charge on the surface of the polymer chains (such as NH_3_^+^ and OH_2_^+^). After adding ionic liquid, the negative ion CH_3_COO^−^ will be attracted by the cations on the chain (Figure 9b) [106]. As a result, the introduction of ionic liquids increases the conductivity of the separator, but also leads to changes in the crystallinity, morphology, and chemical bonds of the separator [106]. The oxygen rich functional groups of graphene can initialize the nuclear deposition sites and have a reversible epitaxial electrodeposition effect. It can also be used to modify the composite cellulose acetate separator to enhance its conductivity [113]. In addition, by optimizing the preparation process of the biological plasma membrane, biomass nanofibers can also be used to successfully construct ultra-thin and ultra-tough separators with dendrite resistance. In addition to the properties of biomass itself, these excellent properties are derived from the high purity cellulose content (close to 100%), large aspect ratio and ultrafine nanofiber network obtained by microbial fermentation. Compared with the GF membrane, the Nafion membrane, polyolefin membrane and ordinary cellulose membrane, the thickness of the biomaterial-based nanofiber membrane is the lowest (as low as 9 mm), but the area density is lower, and the tensile strength is the strongest [18].

## 6. Biomass Used in Binders

The binder can firmly bond the electrode material and its active substances and conductive agents to the current collector, providing a stable interface for the movement of ions and charges, solving a series of problems such as the dissolution of active substances, and giving the battery a longer service life [15,114]. Polyvinylidene fluoride (PVDF) as a common binder has strong electron-withdrawing functional groups and a high dielectric constant, which can improve ion transport, attract positively charged ions such as Zn^2+^, and provide a high concentration of charge carriers. However, the PVDF binder is not environmentally friendly, the organic solvent used is not only flammable and volatile, but also expensive and difficult to recycle [115].

Biomass water-based binder has a low cost, no pollution, no strict processing requirements for air and humidity, and the solvent evaporation is fast. Moreover, the biomass water-based binder has an average swelling trend in the electrolyte. Its strong electron-withdrawing functional groups, carboxymethyl groups and other functional groups can also weaken the hydrogen bond between hydroxyl groups, promote the penetration of the electrolyte, and make the binder have a high degree of wettability and flexibility, which can improve the cycle stability of zinc-ion batteries [116,117]. The mixture of sodium carboxymethyl cellulose, sodium cellulose acetate, sodium alginate, sodium carboxymethyl cellulose and polyvinyl alcohol can be used as binders. The bonding performance of carboxymethyl cellulose is better than that of PVDF, which can be ascribed to the close connection of carboxymethyl cellulose and all components, reducing the influence of the phase change of the active material on the life of the battery (Figure 9c) [10,30,118]. Moreover, carboxymethyl cellulose has the most active sites, the O atoms of carboxylate and hydroxide groups of carboxymethyl cellulose form many binding sites with the Mn atoms of α-MnO_2_. In addition, ZIB/ carboxymethyl cellulose has the best cycle performance (Figure 9d) [10].

Overall, significant progress has been made in this field in recent years. Table 1 summarizes the recent reported biobased ZIBs. Admittedly, the development of biobased ZIBs is still at the primary stage. To move toward the industrialization of biobased ZIBs, more investigative work should be devoted to this field.

## 7. Conclusions and Foresight

Exploring the application of biomass in zinc-ion batteries is of great significance for energy storage and conversion and the development of new applications of biomass. In this review, we comprehensively summarize some problems existing in aqueous zinc-ion batteries and propose strategies to solve these problems by using biomass with different components. However, there is still a long way to go to realize the industrial application of biomass in zinc-ion batteries. The following views are put forward:

(1)Biomass polymers can be directly extracted from biomass. The functional groups of biomass polymers have an important influence on their applications. Grafting modification or modification with active compounds can improve their mechanical properties, hydrophilicity, and flexibility. The structure of biochar plays an important role in its application. The preparation of biochar can inherit the original structure of biomass and can also be restructured. Plants with high cellulose content, such as bamboo, can effectively retain their original structure and system by carbonization. In order to improve the porosity and better retain the original structure, it can be chemically activated to remove impurities. In addition, the template method can be adopted to better control its pore structure. Biological/ice templates can be introduced to effectively reduce the cost, complexity and danger of the template method. Biochar used in the field of energy storage can improve its conductivity by heteroatom doping. Protein, as an important part of all organisms, has been widely used to synthesize nitrogen-containing biochar.(2)The energy density and energy storage effect of biomass-based energy storage devices are much lower than those of traditional metal-based energy storage devices and more efforts are needed to increase conductivity. The key problem to be solved when biomass and its derived materials are used as electrode materials is that biomass itself is not conductive and has limited functions; thus, biomass materials are usually prepared into carbon materials for battery electrodes or conductive additives. In addition, heteroatoms can also effectively increase the active sites and improve the coulombic efficiency of the electrode. The introduction of biomass molecules with redox active groups, the application of dynamic and adaptive coatings, and the design of complex three-dimensional hierarchical porous structures are all effective control strategies to construct advanced biomass materials for zinc-ion battery electrodes.(3)According to the unique features and characteristics of biomass, more efforts should be devoted to achieving the organic combination of biomass and ZIBs components. For example, a gel electrolyte can be obtained by adding biomass materials to the electrolyte, which can effectively inhibit the growth of zinc dendrites and the passivation of zinc-ion battery anode. However, the application of biomass in the electrolyte leads to the lower stability of zinc-ion batteries; in addition, the wide temperature range is also a challenge for its commercial application. Antifreeze compounds, such as ethylene glycol and Zn(ClO_4_)_2_, can be applied to reduce the freezing point of water electrolyte. The copolymerization of biomass and PAM can enrich the network structure of the electrolyte, improve the conductivity, and accelerate the commercialization of the biomass electrolyte.(4)The application of water-based biomass in the separator focuses on controlling the thickness and porosity of the separator. Modification with biomass is an excellent method to regulate the thickness and porosity of the separator. The modification of commercial glass fiber physical separator by biomass can improve its hydrophilicity and flexibility, increase the porosity and ion exchange rate of Nafion membrane. However, the organic combination of low cost, high ion exchange rate, excellent mechanical properties and toughness is still a challenge for biomass-derived zinc-ion battery separators. In addition, some key drawbacks of adhesives must be overcome, such as variable mass, weak adhesion, and hardness at low temperatures. Biomass, including sodium carboxymethyl cellulose and sodium alginate, have good water-based adhesion, and their high solubility still needs to be copolymerized with polymers.

## Figures and Tables

**Figure 1 molecules-28-02436-f001:**
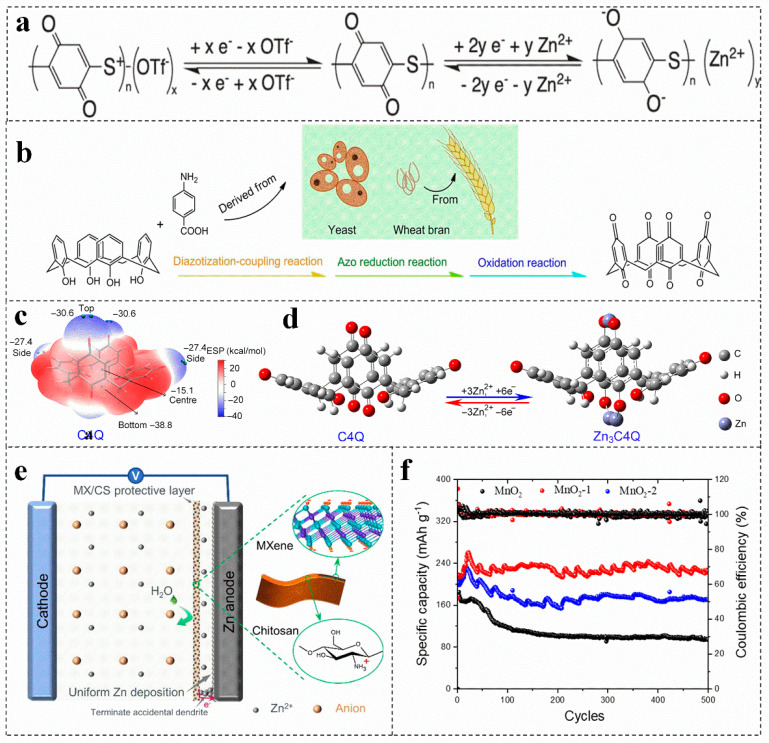
(**a**) The reaction mechanism of quinone electrode. Reproduced with permission [54]. Copyright 2020, Electrochemical Society, Inc. (**b**) Schematic diagram of preparing calix[4] quinone, (**c**) the ESP- mapped molecular van der Waals surface of calix[4] quinone, (**d**) optimized configurations of calix[4] quinone before and after Zn ion uptake. Reproduced with permission [39]. Copyright 2018, American Association for the Advancement of Science. (**e**) MXene/chitosan mechanism of action. Reproduced with permission [38]. Copyright 2022, Elsevier. (**f**) Cycling performances of different batteries. Reproduced with permission [57]. Copyright 2021, MDPI (Basel, Switzerland).

**Figure 2 molecules-28-02436-f002:**
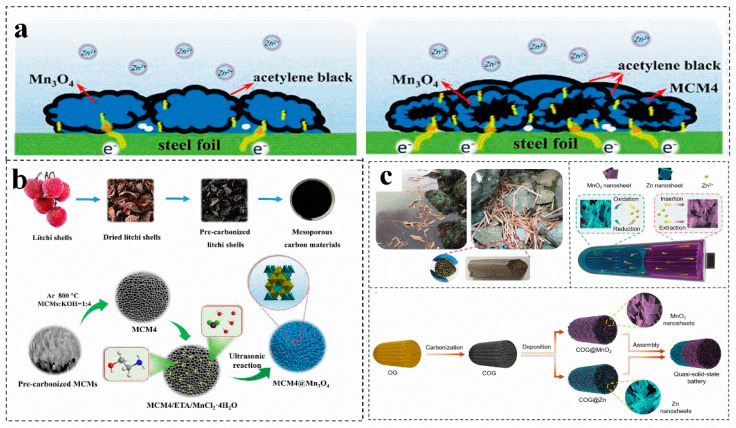
(**a**) Schematic diagram of electronic transmission on pure Mn_3_O_4_ and carbon materials@Mn_3_O_4_ electrodes. (**b**) Schematic illustration for the synthesis of carbon materials@Mn_3_O_4_. (**a**,**b**) Reproduced with permission [62]. Copyright 2020, Elsevier Ltd. (**c**) Schematic illustration of the carbon materials fabrication. Reproduced with permission [63]. Copyright 2020, Royal Society of Chemistry.

**Figure 3 molecules-28-02436-f003:**
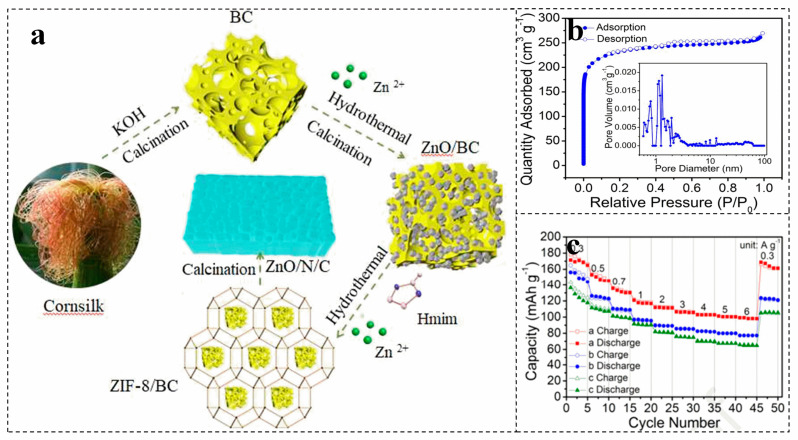
The formation mechanism (**a**), N_2_ adsorption–desorption isotherms and pore size distribution curve (**b**), and rate performance (**c**) of honeycomb ZnO/N/C cathode. (**a**–**c**) Reproduced with permission [67]. Copyright 2021, Elsevier.

**Figure 4 molecules-28-02436-f004:**
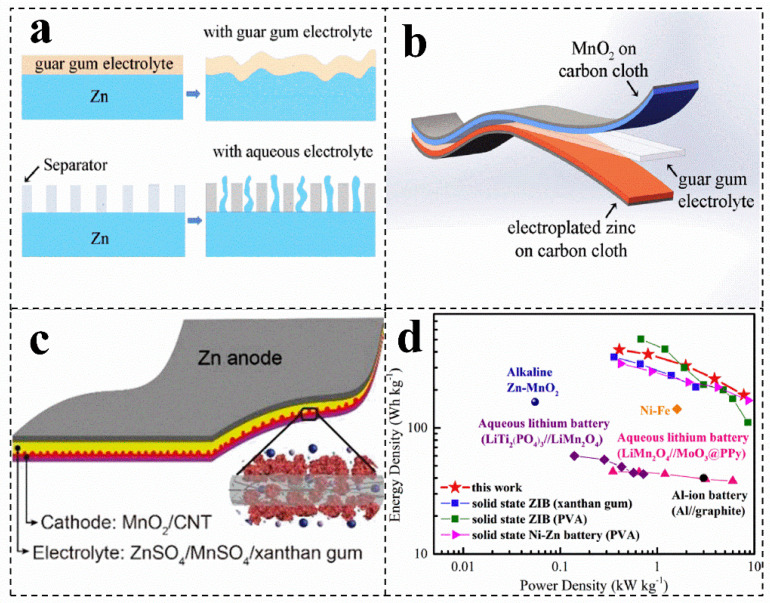
(**a**) Schematic illustration of the structure of the solid-state Zn-MnO_2_ battery. (**b**) Schematic diagram of the morphology change of the zinc foil in the guar gum electrolyte and aqueous electrolyte after cycling. (**a**,**b**) Reproduced with permission [8]. Copyright 2019, Elsevier. (**c**) Schematic showing the structure of the gum Zn-MnO_2_ battery. Reproduced with permission [71]. Copyright 2018, Royal Society of Chemistry. (**d**) Ragone plots of the solid-state ZIBs with guar gum electrolyte. Reproduced with permission [8]. Copyright 2019, Elsevier.

**Figure 5 molecules-28-02436-f005:**
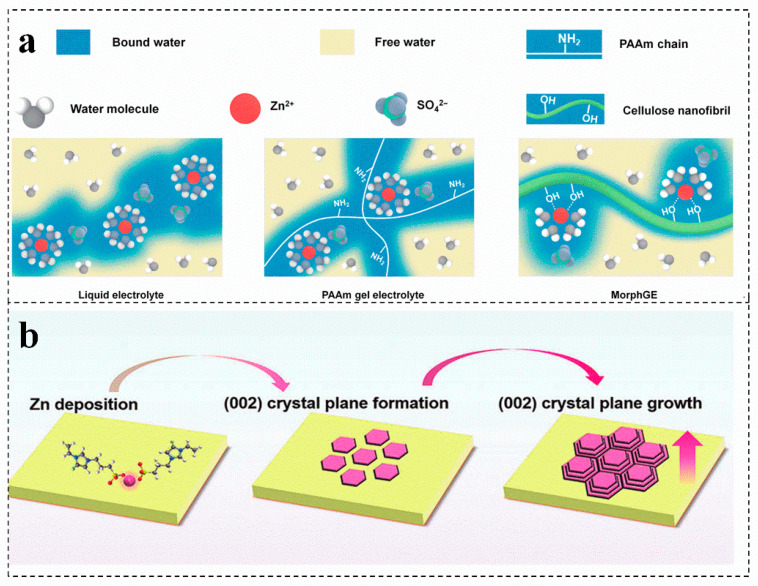
(**a**) Schematic diagrams of the solvation structures of Zn ions in different electrolytes. Reproduced with permission [79]. Copyright 2021, Tsinghua University Press. (**b**) Schematic illustration of the evolution of Zn (002) crystal plane during plating/stripping process. Reproduced with permission [7]. Copyright 2022, Wiley-VCH.

**Figure 6 molecules-28-02436-f006:**
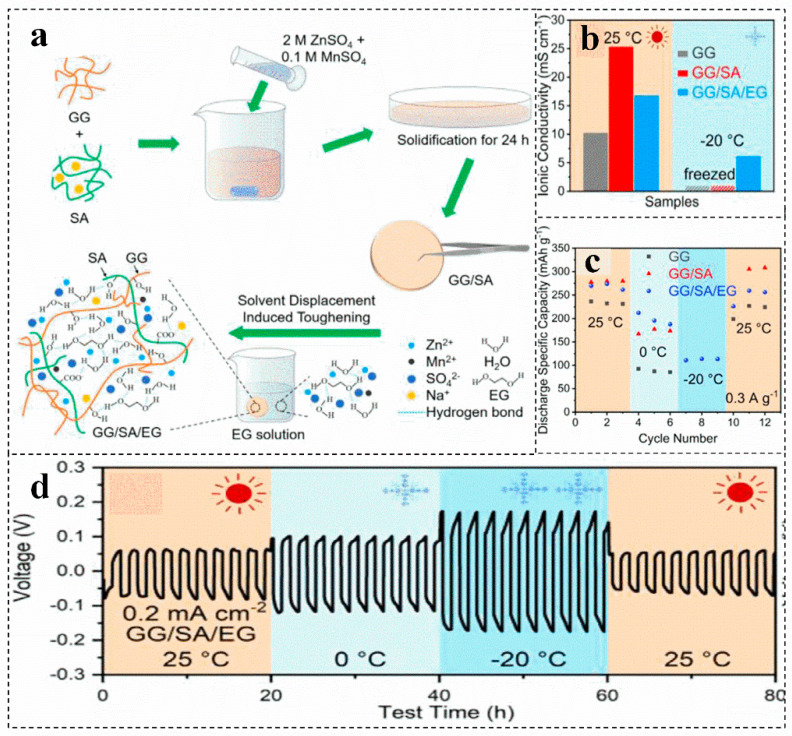
Schematic illustration of the fabrication (**a**), comparison of the ionic conductivity (**b**), discharge capacities of zinc-ion battery (**c**), and voltage profile of the Zn/Zn symmetrical cell (**d**) of different electrolytes at 25 °C and −20 °C. Reproduced with permission [85]. Copyright 2021, Elsevier.

**Figure 7 molecules-28-02436-f007:**
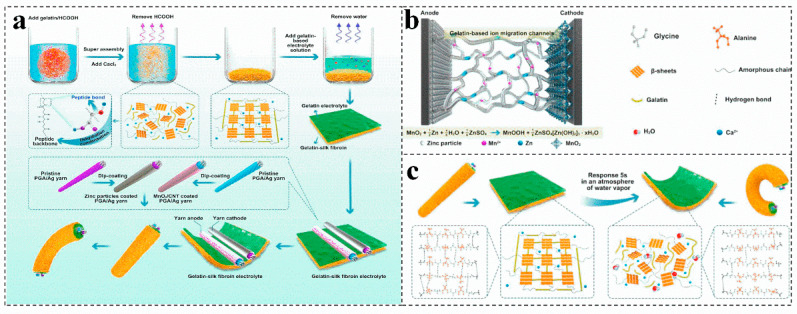
Schematic diagram of fabrication and encapsulation of the fiber-shaped (**a**), schematic of the gelatin-silk fibroin electrolyte (**b**), and the shape-controllable performance mechanism diagram (**c**), in a transient zinc-ion battery. Reproduced with permission [73]. Copyright 2021, Elsevier.

**Figure 8 molecules-28-02436-f008:**
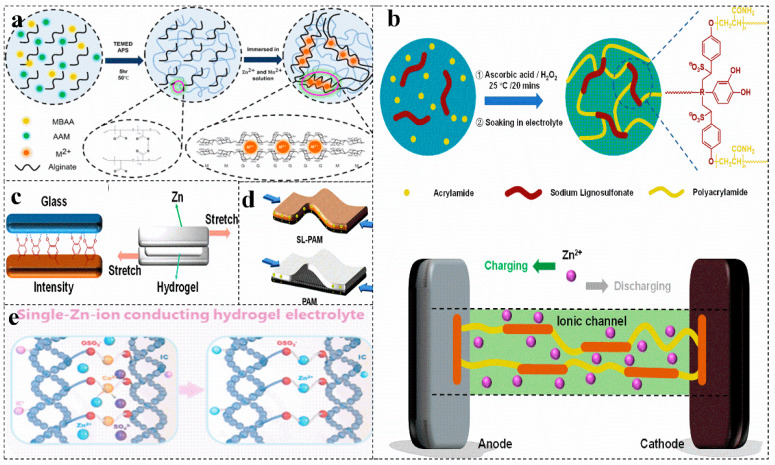
(**a**) Schematic diagram of the synthetic route of the hybrid polymer electrolyte sodium alginate-PAM. Reproduced with permission [93]. Copyright 2021, American Chemical Society. Schematic illustration of synthesis (**b**) and mechanism of strong adhesion to glass and evaluation of the viscosity (**c**) and diagram of their charge transfer in the bent state (**d**) of different electrolytes in zinc-ion battery. Reproduced with permission [96]. Copyright 2021, Academic Press Inc. (**e**) Schematic of the ion exchange process to prepare PAM-iota carrageenan electrolyte. Reproduced with permission [56]. Copyright 2021, American Chemical Society.

**Figure 9 molecules-28-02436-f009:**
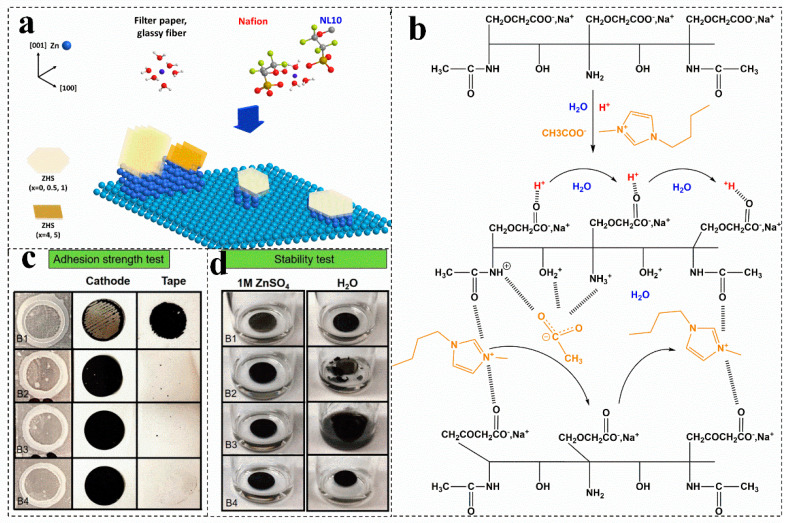
(**a**) Proposed scheme to summarize the effects of membranes on the surface of Zn metal. Reproduced with permission [9]. Copyright 2019, Wiley-VCH Verlag. (**b**) Mechanism of ion mobility in the carboxymethyl chitin ionic liquid-based membrane. Reproduced with permission [106]. Copyright 2021, Elsevier Ltd. Adhesion strength (**c**) and stability tests (**d**) of electrolytic manganese dioxide electrodes with different binders. Reproduced with permission [10]. Copyright 2021, Wiley.

**Table 1 molecules-28-02436-t001:** Summary of the application of biomass in various components of zinc-ion batteries and the structure and electrochemical performance of batteries in recent years.

Cathode Material	Anode Material	Device Configuration	Electrolyte	Capacity (mAh g^−1^)	Energy Density (Wh kg^−1^)	Power Density (kW kg^−1^)	Cycling Stability	Conductivity (mS cm^−1^)	Features	Ref
COG@MnO_2_	COG@Zn	Sandwich structure	ZnSO_4_/MnSO_4_	369.73	420.1	/	95.2% capacity retention after 3000 cycles	/	Zn^2+^ insertion/ extraction	[63]
MCM4@Mn_3_O_4_	Zn foil	Sandwich structure	ZnSO_4_/MnSO_4_	275	/	/	80% capacity retention after 2000 cycles (0.6A g^−1^)	/	Zn^2+^ insertion/ extraction	[62]
α-MnO_2_/agar	Zn foil	Sandwich structure	ZnSO_4_/ MnSO_4_	384.7 (0.25 A g^−1^)	/	/	85.6% capacity retention after 500 cycles (0.25 A g^−1^)	/	Zn^2+^ insertion/ extraction	[57]
C@V_2_O_5_	Zn film	Sandwich structure	Zn(CF_3_SO_3_)_2_	361 (0.5 A g^−1^)	/	/	71% capacity retention after 2000 cycles	/	Zn^2+^ insertion/ extraction	[43]
CQ4	Zn foil	Sandwich structure	Zn(CF_3_SO_3_)_2_	335	220	/	87% capacity retention after 1000 cycles (0.5 A g^−1^)	/	Zn^2+^ insertion/ extraction	[39]
ZnO/N/C	Zn plate	Sandwich structure	ZnSO_4_	172.2 (0.3 A g^−1^)	112.8	2.9	97% capacity retention after 8000 cycles (0.3 A g^−1^)	/	Zn^2+^ insertion/ extraction	[67]
CH-EMS- MnO_2_	Zn	Sandwich structure	ZnSO_4_/ MnSO_4_	415 (0.5 A g^−1^)	/	/	90% capacity retention after 1000 cycles (0.5 A g^−1^)	/	Zn^2+^ insertion/ extraction	[119]
Gelatin- MnO_2_	Zn	Sandwich structure	ZnSO_4_/ MnSO_4_	330 (0.5 A g^−1^)	/	/	80% capacity retention after 1000 cycles (0.5 A g^−1^)	/	Zn^2+^ insertion/ extraction	[61]
N/E-HPC-900	Zn foil	Sandwich structure	KOH/Zn(Ac)_2_	801	955		/	/	Zn^2+^ insertion/ extraction	[120]
MnO_2_/CNT	Zn foil	cable-type	GSF	311.7	/	/	94.6% capacity retention over 100 cycles	/	high capacity retention of 82.5% after 80 bends	[73]
Na_0.65_Mn_2_O_4_	Zinc paste	Sandwich structure	SA-PAM	160	/	/	96% capacity retention 2000 cycles (2 A g^−1^)	/	Zn^2+^ insertion/ extraction	[93]
MnO_2_/rGO	electroplated zinc	Sandwich structure	ZnSO_4_/MnSO_4_/guar gum	308.2 (0.3 A g^−1^)	416	7.8	100% capacity retention after 1900 cycles	10.7	81.3% capacity retention after continuously bending to 180°for 1000 cycles	[8]
V_2_O_5_/CNT	Zn foil	Sandwich structure	P(ICZn-AAM) SIHE	266.9 (0.3 A g^−1^)	/	/	110% capacity retention after 150 cycles	21.5	/	[56]
MnO_2_ /Super P/PVDF	Zn foil	Sandwich structure	GG/SA/EG	354.9 (0.15 A g^−1^)	432.2	7.43	91.52% capacity retention after 1000 cycles.	6.19	79.5% capacity retention after for 1000 cycles (a radius of 10 mm) (0.3 A g^−1^)	[17]
fibrous PANI	fibrous metal Zn	cable-type	cellulose yarn	152.2	153.2	0.16	/	/	Zn^2+^ insertion/ extraction	[70]
MnO_2_	Zn	Sandwich structure	SL-PAM	/	/	/	/	31.1	Zn^2+^ insertion/ extraction	[5]
MnO_2_	Zn	Sandwich structure	PZIB	120.6	/	/	/	21.88	Zn^2+^ insertion/ extraction	[7]
Zn foil	Zn foil	Sandwich structure	CMChit_IL1_40	/	/	/	/	0.1	Zn^2+^ insertion/ extraction	[106]
β-MnO_2_	Zn foil	Sandwich structure	ZnSO_4_ (Nafion mem- branes separator)	236 (0.3 A g^−1^)	/	/	/	/	Zn^2+^ insertion/ extraction	[9]
ZnMn_1.71_O_4_	Zinc foil	Sandwich structure	ZnSO_4_/MnSO_4_ (g-AGM separator)	103	/	/	91.3% capacity retention after 500 cycles,	/	Zn^2+^ insertion/ extraction	[63]
MnO_2_/CNT	Zn-P	Sandwich structure	ZnSO_4_/MnSO_4_ (CT separator)	/	/	/	/	/	Zn^2+^ insertion/ extraction	[121]
MnO_2_	Zn	Sandwich structure	ZnSO_4_ (BCM separator)	173.8 (0.6 A g^−1^)	/	/	/	/	Zn^2+^ insertion/ extraction	[18]
Hydrated VO_2_/CC	Zn	Sandwich structure	Zn(CF_3_SO_3_)_2_ (rGO/CA separator)	/	/	/	/	/	Zn^2+^ insertion/ extraction	[113]
α-MnO_2_	Zn foil	Sandwich structure	ZnSO_4_/MnSO_4_ (CMC binder)	93 (6 A g^−1^)	/	/	73.6% capacity retention after 100 cycles	/	Zn^2+^ insertion/ extraction	[11]
MnO_2_	Zn foil	Sandwich structure	H_2_O (CMC binder)	288 (0.1 A g^−1^)	/	/	/	/	Zn^2+^ insertion/ extraction	[10]

COG: carbonized ocean garbage [63]. MCMs: Biomass-derived mesoporous carbon materials; MCM4: (W_MCMs_:W_KOH_ = 1:4) [62]. CQ4: calix[4] quinone [39]. CH-EMS-MnO_2_: Collagen hydrolysate-electrode microskin-MnO_2_ [119]. N/HPC: N-doped carbons [120]. CNT: carbon nanotubes, GSF: gelatin-silk fibroin [73]. SA-PAM: sodium alginate polyacrylamide [93]. rGO: reduced graphene oxide [8]. SIHE: single-Zn-ion conducting hydrogel electrolyte; P: ICZn-Aam (synthe-sized with IC and Aam); IC: iota carrageenan; Aam: acrylamide; CNT: carbon nanotubes [56]. PVDF: polyvinylidene fluoride; GG: guar gum; SA: sodium alginate; EG: ethylene glycol [17]. PANI: polyaniline [70]. SL-PAM: sodium lignosulfonate-polyacrylamide [5]. PZIB: bacterial cellulose as gel electrolyte [7]. CMChitin: Carboxymethyl chitin; BMIM[Ac]: 1-Butyl-3-methylimidazolium acetate; IL: ionic liquid [106]. G-AGM: gelatin modified AGM; AGM: glass fibers with the diameter in a range of 10nm–2μm [63]. CNT: carbon nanotubes; Zn-P: Zn Powder Electrode; CT: cotton towel; HP: hydrophilic polyolefin [121]. GF: Glass microfiber; CA: cellulose acetate; CC: carbon cloth [113]. BCM: bacterial cellulose membrane [18]. CMC: sodium carboxymethyl cellulose [10,11].

## Data Availability

Not applicable.

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
