# Peer review of "Application of Biomass Materials in Zinc-Ion Batteries"

_molecules, 2023, doi:10.3390/molecules28062436_

Round 1

Reviewer 1 Report

Reviewing the manuscript, I find it interesting and believe it may be published in this journal after the authors have checked some suggestions. Check the references position. They are after the sentence point. This creates confusion, because it is not clear whether they belong to the beginning or ending sentence. Example: Quinones present completely insoluble in water, which have a variety of properties when applied to zinc ion batteries.[30]. As it is a review work, more details and explanations must be provided for each figure presented. For example, Figure 1a should be discussed in detail. As well as the other figures of the work. The figures must be distributed or separated so that the details can be visualized. Many clustered figures do not allow to observe in detail the information that is presented. About the methods for obtaining biomass, what can the authors include or suggest in their review to guide researchers working in the area?

Reviewer 2 Report

The review for biomass utilization in zinc battery application is rather extensive and complete. To improve the review it is best to include a subtopic on generality of zinc ion battery i.e the principle, mechanisms, the range of energy density, what are main components etc. Then only to put examples of usage of biomass as different components in battery. 

In table 1, the label on conductivity but the content in the column is rather suits as feature as there are no values presented. 

Round 2

Reviewer 1 Report

This version may be considered for publication.